**Title**

"*To Detain Is to Take*" – the carceral mobilities of Australia's maritime migration governance

**Abstract**

Over the past four decades, maritime geographies have become prominent sites of policing and containing human migration. While there is important scholarly work on these contested liquid geographies charting the changing techniques of migration control, what continues to demand attention is the containment that is achieved through systemically keeping migrants mobile at sea. This paper explores maritime migration governance in Australia, examining the coerced mobilities that follow interdictions at sea and their carceral nature. I interrogate the High Court case, *CPCF v Minister for Immigration and Border Protection*, which addresses the extended detention of 157 Tamil asylum seekers at sea in June 2014. Through analysing the language used in this case, such as the conclusion by the majority that "to detain" a migrant at sea mandates a concomitant duty "to take" that migrants somewhere else, I highlight how coerced mobility has become central to Australia's strategy of migration governance and the indefinite detention at sea that this has come to legitimate. This will reveal the extent to which carcerality informs migration governance in Australia's maritime geographies.

**Introduction**

In late June 2014, an Indian flagged vessel carrying 157 Tamil asylum seekers from Sri Lanka went into distress in the maritime geography off Australia after an oil leak caused a fire in the engine room. On June 29, after contacting the Australian Maritime Safety Authority, the vessel was intercepted by The Royal Australian Navy and the Customs and Border Protection Service and the asylum seekers were boarded on an Australian Customs vessel. The boat was intercepted 16nm off Christmas Island, outside Australia's territorial waters, yet within the state's contiguous zone. It was decided by Australia's National Security Committee of Cabinet that the asylum seekers should be returned to India, as it had been a transit country in the journey of these asylum seekers. In a ten-day period between July 1 and 10, the vessel upon which the asylum seekers were being held sailed toward India. As no former agreement had been made to disembark the migrants in India, the Customs vessel was forced to remain offshore while negotiations between India and Australia proceeded. No agreement was reached and after nearly a month detained at sea, the asylum

seekers were disembarked at the Australian territory of the Cocos (Keeling) Islands, before being transferred to the Curtin Immigration Detention Centre in West Australia, and then taken to Australia's offshore immigration detention centre in Nauru. Several months later, this ad hoc practice of forced mobility and prolonged detention at sea was rendered lawful in the High Court of Australia, with the case *CPCF v. Minister for Immigration and Border Protection*. The presiding judges ruled that it was a lawful form of detention, in particular claiming that "to detain" a migrant at sea mandates an obligation "to take" that migrant somewhere. As Justice Gageler notes, "Detention of a person under the provision triggers a concomitant duty to take the detained person to a place.".[1] As is explored in this article, the term "taking" is specifically impactful as the destination of such journeys was left resoundingly vague. The judges further ruled that there cannot be a constriction on the time that is needed to facilitate this "taking", legitimating indefinite detention at sea for the purpose of realising such journeys. As Australia exists as the international example of an offshore migration governance regime that has managed to "successfully stop" migrant arrivals, this case and the at-sea strategies it legitimates holds profound significance internationally.

This article examines migration control in maritime geographies in Australia, asking the questions: how is maritime migration governance increasingly carceral and what role does the sea play in justifying this carcerality? I perform a discourse/materiality analysis of the High Court case *CPFC v Minister for Immigration and Border Protection* to examine the legitimation of strategies of detention and coerced mobility at sea. This case is specifically profound as it took place in Australia's High Court, the nation's apex court responsible for clarifying the application of domestic law and policy. The court case specifically justified detention at sea as requisite and necessary to a strategy of migration governance, with the sea framed as demanding indefinite confinement. While Australia has been reliant on coerced mobilities at sea for several decades, this case established the maritime environment of Australia to be an explicitly carceral space for migrants. This article begins with an analysis of migration governance and the need to examine it through a lens of carcerality. It then examines the growing importance of coerced mobilities at sea in Australia's agenda of

---

[1] Para 376. *CPCF v Minister for Immigration and Border Protection* [2015] HCA 1 28 January 2015 S169/2014.

maritime migration control, before interrogating how the *CPCF* case legitimated indefinite detention and myriad coerced mobilities at sea.

**The Sea as a Carceral Geography for Migrants**

There is a great deal of research investigating the changing technologies and techniques of policing migration at sea (i.a. Carrera and Den Hertog, 2015; Cuttitta, 2018a; Den Hertog, 2012; Tazzioli, 2018); the evolving geographies of maritime migration governance (i.a. Basaran, 2010; Bialasiewicz, 2012; Everuss, 2020; Hyndman and Mountz, 2008); and the violence of such agendas (i.a. Mazzara, 2019; Mirto, 2018; Presti, 2019; Squire, 2017; Stierl, 2017). Such research expands knowledge on how borders function at sea and challenges where we understand them to be. Through such work, maritime borders have come to be understood as unfixed from certain geographies, appearing in shifting locations in order restrict the movement of mobile populations rendered "undesirable". The increasing digitalization of migration governance has further exaggerated the mutability of borders, with security technologies and networks of data exchange allowing these "undesirable" mobile populations to be identified and tracked across terraqueous geographies (i.a. Broeders, 2007; Pallister-Wilkins, 2011; Tazzioli, 2018). Despite the amorphous nature of contemporary borders and the expansive literature documenting the evolving ways they manifest, what the term "border" implies across its various applications remains consistent: the delineation between an "inside" and an "outside" and the methods of policing and securitising that perform a "keeping out". In this article, I move beyond border language in order to focus on mobility as a method of governance, examining how migrants are contained through being kept mobile at sea. Within this analysis of mobility as governance, there is no clear inside or outside, thus complicating the application of the term "border". Moreover, rather than limiting migratory journeys, at-sea mobilities works to redirect, prolong, and extend journeys. Focusing on mobility at sea as a method of governance thus draws to the fore unique aspects of the migration governance agenda and the ways it works in maritime geographies.

This close attention to mobility draws on recent work in security and migration studies. In particular, Huysmans (2021, p. 6) has articulated the significance of "giving primacy to movement" (2021), articulating how "conceptualizing life as motion without stasis invites distinct analytics of security". As Huysmans emphasises, this focus on motion encourages "letting go of defining the politics of security as an enactment of continuously dividing

insides and outsides" (Huysmans, 2021, p. 7). Tazzioli (2019) examines this in the context of migration governance within Europe, drawing emphasis to migratory mobilities, rather than the immobility implied by borders. Tazzioli (2019, p. 129) details "mobility as a technique for neutralising and dividing emergent collective formations" of migrants populations, contributing to dispersals of people. Thus, by directing focus to mobilities we are persuaded to engage with geographies such as the maritime as defined by series of encounters which lead to onward mobilities, examining the control and containment that such mobilities reflect. This article hence builds on the nascent work examining the significance of mobility as a method of governance to demonstrate its centrality to migration governance at sea. As is explored below, these mobilities are understood as carceral in nature, due to their coerced nature and the containment they come to enforce.

Coerced mobilities on behalf of governing agents at sea are often interrogated as "pushbacks" or "returns" (i.a. Borelli and Stanford, 2014; Cuttitta, 2018b). "Return" refers to a strategy of taking maritime migrants back to the territory from which they departed, without providing an asylum process. The Australian government has, for decades, facilitated "returns" via the sea, as have many Southern European/Mediterranean states who return migrants to departure countries including Libya and Tunisia, as well the US government which has returned migrants via the sea to Cuba, Haiti, and beyond. While often used interchangeably with the term "return", "pushback" is distinct in that it refers to a strategy of expelling migrants immediately after they have crossed a border, removing them from territorial waters and taking them to the high seas or to the territorial waters of third-party states. This thus denotes a practice not of taking migrants to a port or a grounded geography, but rather leaving them at sea. This strategy of maritime governance has been well-documented in the context of Greece and the pushback of boats at sea to Turkish waters, where Greek authorities often leave boats without motors, ensuring they are unable to attempt to re-enter Greece (Enkelejda, Koka; Denard, 2020; Human Rights Watch, 2020; Pro Asyl, 2013). There are also well-documented strategies of pushbacks executed by Italian, Maltese, and Spanish authorities, where they remove migrants to maritime geographies outside their own territorial waters (Amnesty International, 2020; Tondo, 2021). "Pullbacks" also transpire in the Mediterranean, in which the training and funding of third-party state coast guards lead to migrant vessels being pulled-back to the Southern Mediterranean coasts (Carrera and Cortinovis, 2019). While the terms "return", "pushback" and "pullback" are relevant to this study, each of these terms implicitly affirms the significance of boundaries at sea which

migrants either cross, are prevented from crossing, or taken back across. They thus reify the idea of the maritime as defined by clear and determined lines at sea. My focus is not on *crossing* or *returning* but rather on the centrality of mobility to governance. I thus use the term "coerced" mobility to emphasise the myriad enforced mobilities that take place at sea, within which the concepts of "inside" and "outside" lose significance.

This article builds on previous research investigating the hollowing out of rights for migrants in Australia's maritime geography, through which governments have provided a framework to enable carceral practices at sea (Dickson, 2021). The geography of the maritime has a variant relationship to the sovereign territory, at times framed as part of the sovereign territory of the state, while at other times rendered distinctly beyond the state (Peters, 2011). As has been explored, this variant relationship has facilitated states "hollowing out" rights for migrants, expunging the applicability of articles of the Refugee Convention or the Convention at large from their maritime geographies (Dickson, 2021). At the same time, they expand migration governance agendas in these same geographies, leading to a de-territorialising and re-territorialising of the sea. This was demonstrated in the US in 1993 through the Supreme Court case *Sale v Haitian Councils Centres inc*, in which the Refugee Convention was ruled not to apply at sea. As a result, the US government could police, intercept, and return Haitian maritime arrivals without providing them access to an asylum procedure. Australia extrapolated on this to eliminate their migration zone for those arriving in an "irregular" manner, meaning migrants cannot claim asylum. Due to Australia's islanded geography and regional agreements, irregular travel is almost exclusively constricted to maritime arrivals (Dickson 2021, p. 7). Through these changes to the legal geographies of the maritime, this liquid space has been rendered a blurry sovereign geography that is at once outside and inside the state in the context of migration governance; beyond a geography of rights for migrants yet within a geography of policing. As a result, the sea has become a "carceral wet" space for migrants, where they are confined to a condition of policing in abstraction of rights (Dickson 2021). This article extrapolates on the legal changes which have given way to the carceral potential of the sea to detail how this carcerality manifests through the proliferation of coerced mobilities and at sea detentions in maritime migration governance.

Using carcerality to interrogate these at sea mobilities reveals three distinct aspects of this strategy of migration governance. On the first hand, it highlights the significance of mobility

as a method of containment. The carceral is not premised on the suspension of a person's mobility but rather on the control over mobility, in fact, coerced mobilities are central to the design of containing and separating people (Martin and Mitchelson, 2009; Mincke, 2020; Moran, 2016; Moran, Turner and Schliehe, 2018; Turner and Peters, 2017). Keeping migrants mobile is thus exposed to be far more than an incidental aspect of migration governance; instead, it emerges as integral to the design of control and containment. Secondly, it illuminates how this mobility is harmful and has a disciplinary intention. Coerced mobility at sea is part of a logic of migration governance, in which states attempt to impede migrants from reaching the state's territory through keeping them mobile and thus unable to claim rights. As a result, this mobility has an "intentional" and "detrimental" effect as it functions as part of the apparatus of regulating migration and discouraging future mobilities (Moran, Turner and Schliehe, 2018, p. 678). Finally, these mobilities keep migrants under the governing powers of the state, while holding them in a condition of reduced rights, suspending them in a deeply carceral circumstance. While in more formal carceral sites, such as prisons, detained persons are not entirely beyond legal frameworks, Brown (2014, p. 177) highlights how prisoners, like refugees and other detainees, "share restricted rights and weaker claims to citizenship". Using carcerality to consider forced mobilities at sea thus forces us to examine the significance of these mobilities to the control and exclusion of migrants in maritime geographies.

To explore the centrality of carceral mobility to Australia's governance of maritime migration I draw on a methodological approach which privileges both discourse and materiality. As Aradua et al, (2015, p. 62) write, "matter is not inert nor the passive end product of discourses but an active factor in the construction of relationships in discursive-material processes; it actively shapes how subjects and objects of insecurity are constructed, regulated and materialized in discourse". To consider the "co-constitution of matter and discourse", Aradua and others propose the need to focus on the relationality between discourse and matter to detail how they perpetually reconfigure each other (Aradau *et al.*, 2015, pp. 62–63). In the context of this study, this co-constitution is between a discourse of migration detention and the space of the sea as one that legitimates detention. The research design of this paper is based a discourse/materiality analysis of the transcripts of the High Court proceedings between the plaintiff and defendant, *CPCF v. Minister for Immigration and Border Protection & Anor* Case S169/2014, as well as the response by the presiding judges *CPCF v Minister for Immigration and Border Protection [2015] HCA 2015*. This case

is chosen as a point of analysis for several reasons. Firstly, it refers to a pinnacle in Australian migration governance at sea, where strategies that contravene the Refugee Convention of detention at sea and *refoulement* were ruled to be lawful strategies. While legal scholars have examined the *CPCF* case, considering the framing of the statutory and executive power of the government in this case (Emerton and O'Sullivan, 2015; Marmo and Giannacopoulos, 2017; Tomasi, 2015), there has not been consideration of what this case means for the framing of the maritime as a space which permits detention and coerced mobility and the way in which the sea was used to legitimate such practices. Secondly, as the primary institution in Australia codifying how policy should be applied, this High Court case was profound in contributing to the normalising of a discourse on the sea as a geography that necessitates detention and onward mobilities. The research design relied upon a coding of the two transcripts, identifying references to detention at sea as well as the concept of mobility at sea. In this paper, I paid greater attention to the transcript of the ruling by the judges due to their statements having conclusive influence determining the implementation of the Maritime Powers Act (2014) and legitimating strategies of migration control at sea. The article further relies upon an analysis of preceding migration policies to illustrate the significance of mobility to the Australian migration governance agenda.

**Australia and Coerced Mobility at Sea**

The coerced mobility at sea that has come to inform migration governance in Australian maritime geographies is not without context. As a settler convict state, Australia has a very recent history shaped by administering punishment through coerced maritime mobilities. Beginning in 1788, the transportation of convicts to the Australian territory lasted 80 years, with the final convict transportation vessel landing in West Australia in 1868, a little over 150 years ago. During this time 168,000 prisoners were transported from the UK to Australia (Godfrey and Williams, 2018). In addition to this transportation of convicts to Australia, from 1788 until 1901, the colony also relied upon a network of at least eleven offshore carceral islands (Roscoe, 2018). These carceral islands were spaces of secondary punishment for re-offending convicts as well as spaces to separate and contain indigenous Australians that resisted colonialization, displacing them from their land and distancing them from the mainland colony. As Roscoe (2018, p. 48) writes, "'punitive relocation' to offshore islands was an important part of the colonial system of punishment that emerged in Australia". The Australian settler colony relied upon a maritime carceral mobility, from the transportation of convicts to the colony, to the governing and policing of persons within the colony. Maritime

mobility and a distance maintained through seascapes was thus central to the governance of disobedient bodies in Australia. This very recent history is not irrelevant to the formal political discourse in Australia that instructs that detaining a migrant at sea necessitates a taking of that migrant somewhere else, somewhere beyond mainland Australia.

Coerced mobility has, over the last three decades, become central to Australia's migration governance. The proliferation of this coerced mobility has been facilitated by the specific geopolitical condition of this maritime environment. Australian maritime migration governance occurs in an exceptionally vast maritime geography, with Australia having an exclusive economic zone at sea that is the third largest in the world. Compared to other maritime regions of mobility control, such as the Mediterranean, there is an absence of private, commercial, and humanitarian actors engaging in migrant rescues. As a result, Australian authorities are the primary actors interacting with migrants at sea, affording the government exclusive authority over what happens to those who are interdicted. Almost all migrant vessels that seek to reach Australia are "detected either en route or upon arrival" (Pickering, 2014, p. 191). Following this interdiction, asylum seekers are removed to various geographies beyond Australia. In other words, they always face onward mobilities. Since 2013 and the commencement of the militarised operation, Operation Sovereign Borders, there has been very little transparency into events at sea (Munns, 2019). Despite recursive reports of human rights abuses at sea, Australia has emerged as an international example of a state that has managed to "stop" maritime migrants yet, as demonstrated below, this is not achieved through rendering migrants immobile, but rather through keeping them mobile.

The coerced mobilities of Australian maritime migration governance developed in a noteworthy way in 2001. This year marked the commencement of Australia's Pacific Solution (2001-2007), a regional policy premised on various offshore sites of interception that aimed to prevent maritime migrants from reaching the territory of Australia and claiming asylum. To realise this, the Australian government bolstered operations at sea which saw migrant journeys interdicted at increasingly distant geographies, at which point migrants where either encouraged to return to their port of departure or, if this was not possible, they were removed to sites of offshore immigration detention, namely those situated on Christmas Island, Manus Island, and Nauru (Loyd and Mountz, 2014, p. 28; White, 2014, p. 9). The analysis of these offshored sites of detention and the way they suspend migrants in geographies that lack both legal accountability and fair access to asylum is beyond the scope

of this article (see i.a. Mountz, 2011; Taylor, 2005; Wallis and Dalsgaard, 2016; Warbrooke, 2014). Instead, I examine that which has received far less scrutiny: the containment and exclusion that is realised through these at sea mobilities that commenced with the Pacific Solution and were later emboldened by successive operations.

Under the Pacific Solution there were three successive operations designed to interdict and remove migrant vessels at sea: Operation Relex (2001-2002), Operation Relex II (2002-2006), and Operation Resolute (2006-ongoing). The commencement of the first Operation Relex signified the beginning of an explicit programme of pre-emptive policing at sea that had a design of interdicting maritime migrants in order to keep them mobile and outside the Australian territory. Within Australian law, intercepted boats carrying migrants are referred to as "suspected illegal/irregular entry vessels" (SIEV). Operation Relex initially had an objective to convince boats to return to Indonesia. However, after the first four "SIEVs" arriving under the Operation could not be "persuaded to return", a practice of "active return" began (Schloenhardt and Craig, 2015, p. 538), which was referred to in government as "the tow-back policy" (Howard, 2003, p. 41). In referring to the development of the practice of enforced returns, the Select Committee inquiry states:

> From the commencement of Operation Relex on 3 September, the initial policy that we were given to implement was to intercept, board and hold the UAs [unauthorised arrivals] for shipment in sea transport - or air transport, but primarily sea transport - to a country to be designated. With SIEV 5, we received new instructions which were to, where possible, *intercept, board and return* the vessel to Indonesia. (Senate Select Committee, 2002 Chapter 2, para 2.69, emphasis added).

Hence, after only a few interceptions at sea, this operation developed from one in which onward mobility at sea would proceed only after a destination country had been designated, to one which executed immediate forced mobilities. Operation Relex was replaced by Relex II in 2002, which had a concomitant agenda. Within five years, these two operations had collectively interdicted all fourteen vessels which had attempted to reach Australia. Of these fourteen, five vessels were forcibly encouraged back to Indonesia, a country which is not signatory to the Refugee Convention and Protocol and where asylum seekers thus have no right to seek protection. Those that were not returned to Indonesia were removed to sites of

offshore immigration detention.[2] As such, *all* interceptions that transpired in the maritime geographies around Australia during this period thus led to the forced mobility of migrants at sea.

It is important to note that these coerced mobilities that have become central to Australia's strategy of migration governance at sea undermined the Refugee Convention. Article 33 of the Refugee Convention protects against *refoulement* or forced return, stipulating that a mobile person shall not be returned to a frontier or territory where "[their] life or freedom would be threatened on account of [their] race, religion, nationality, membership of a particular social group or political opinion". Migrants interdicted in Australia's maritime peripheries are either removed to Indonesia or to places they are fleeing, such as Sri Lanka. Indonesia is not signatory to the Refugee Convention and thus holds no obligation to protect migrants against what is known as onward or chain-*refoulement*. Returning migrants to Indonesia, as a country through which they transited, is thus recognised as a practice that constitutes *refoulement* (Kaldor Centre for International Refugee Law, 2015b). Beginning in 2012, the Australian government began subjecting Sri Lankan asylum seekers to an "enhanced screening" process at sea, which functions as a truncated asylum application. This controversial practice is widely condemned by international human rights groups as not providing adequate access to an asylum procedure (Refugee Council of Australia, 2021a). Those screened out are often transferred to Sri Lankan authorities at sea, with reports from human rights groups indicating that they are often subsequently detained upon arrival in Colombo (Refugee Council of Australia, 2021a). While migration governance at sea in Australia is heavily obscured, the Refugee Council of Australia report that more than 1,000 Sri Lankans have been returned to Sri Lanka as a result of this policy (Refugee Council of Australia, 2021a). This enhanced screening was expanded to including people fleeing by sea from Vietnam, and the Australian authorities have subsequently returned asylum seekers back to Vietnamese authorities (Refugee Council of Australia, 2021c, p. 3). These practices of removal at sea thus defy the state's obligation to *non-refoulement,* which is not only a core principle of the Refugee Convention, of which Australia is a signatory, but it is recognised as customary international law.

---

[2] There were also some fatalities at sea during this period. Three vessels, named SIEVS IV, VI, and X met with disaster at sea during interdiction and return, with a number of individuals perishing during the disasters.

In late 2013, a new government led by the Conservative Prime Minister, Tony Abbott, was formed. Abbott initiated an expressly militarised programme of maritime migration control termed Operation Sovereign Borders (OSB). Under OSB, the government announced that Australia's borders were "shut" to maritime arrivals. To achieve this, the government intensified the scope of coerced mobility at sea, with far more exhaustive measures taken. Emphasising the centrality of mobility to this programme, the former immigration minister, Scott Morrison, described the new agenda as one premised upon "forcibly repelling" maritime migrants (Hall, 2013). Under this operation, the geographies of operation further expanded in scope, with the Australian authorities crossing the territorial waters of Indonesia's archipelagic state on six occasions (ACBPS, 2014). Indonesia has not agreed to accept returned migrant vessels, and as such, Australian authorities more commonly take migrant vessels to the edges of Indonesia's territorial waters and instruct their onward journey to Indonesia (Refugee Council of Australia, 2021c, p. 3; see also Dastyari and Ghezelbash, 2020). Within the first 18 months of OSB, the government prevented 20 vessels carrying 633 asylum seekers from reaching Australia (Jabour, 2015). As of mid-2021, 38 vessels have been subject to at sea mobilities that culminated in removals under OSB (Refugee Council of Australia, 2021b).

These mobilities that transpire in and importantly beyond Australia's maritime territories often lead to the incarceration of migrants in sites of immigration detention. The incarceration that follows these coerced mobilities at sea is not constricted to Australia's island detention centres. The Australian government funds Indonesian immigration detention centres, contributing to the use of detention in Indonesia as a method of controlling human mobility (Nethery, Rafferty-Brown and Taylor, 2013, p. 96). Indeed, Nethery et al (2013, p. 98) highlight how "Indonesia rarely detained asylum seekers before Australia began actively to encourage it to do so.". Moreover, there is evidence that the asylum seekers returned to Sri Lanka face arrest and incarceration (Doherty, 2016). The removal that transpires at sea can thus be understood as an extenuation of the carceral system, transporting migrants to sites of more enduring confinement. As Pickering (2014, p. 188) writes, the Customs vessels that intercept and remove migrants to various offshore geographies "perform a custodial function following the interception of asylum seeker boats to Christmas Island, mainland Australia or designated offshore processing centres such as Nauru and Papua New Guinea.". These onward mobilities at sea thus reflect a broader programme of containing and separating migrants.

Yet, these vessels are not just carceral due to their connection to broader systems of immigration detention, transporting confined people to larger detention facilities. These coerced mobilities turn the vessels holding asylum seekers and keeping them mobile into carceral spaces in their own right. Moran et al (2018) propose that the carceral emerges in the tension between *detriment, intent* and *spatiality*. This refers to the "lived experiences of harm", which may or may not take the form of punishment; the intention behind this detriment, it is not happenchance but rather imposed by an agent or organization; and the spatial dynamic to the carceral condition (Moran, Turner and Schliehe, 2018, p. 677). Coerced mobility signifies the suspension of autonomous movement to facilitate a control that is realised through keeping a subject mobile, signifying both detriment and intent. After all, the onward mobility that transpires at sea, instructed by a governing power, is disciplinary and punitive; it prevents individuals seeking asylum, removing them to various at sea and offshore geographies as a form of punishment for the "irregular" nature of their travel. Moreover, there is a clear spatiality to these mobilities, as asylum seekers are concealed from sight, distanced from populations, and prevented them from accessing legal aid. This coerced mobility at sea thus comes to constitute a carceral space, detrimentally, intentionally and spatially keeping migrants mobile as a form of carceral containment.

**Indefinite Detention and Myriad Mobilities at Sea**

While Australia has an enduring history of employing coerced mobility as a method of governance at sea, the significance of this maritime mobility reached an acme in 2014, when 157 asylum seekers were detained and kept mobile at sea for the period of one month. Throughout this at-sea detention, the Australian Customs Vessel moved to different at sea destinations, attempting to keep the asylum seekers beyond the territory of Australia through attempting to remove them to various overseas geographies. During this prolonged period of detention at sea, the authorities did not inform the asylum seekers or the Australian public where they were located or what their intended destination was. We now know the journey from the point of interception to the coast of India took ten days, and after negotiations failed, the Customs vessel then returned back to the Cocos Islands. The journey across this vast maritime geography, throughout which time the asylum seekers were detained, was an explicit coerced mobility. In transporting the Tamil asylum seekers across this immense maritime geography, they were contained in a two-fold manner: they were contained within

the structure of the Customs vessel, as well as the unbridgeable maritime geographies within which they were secretly transported.

Not long after this event at sea, the Australian government passed the the *Migration and Maritime Powers Legislation Amendment (Resolving the Asylum Legacy Caseload) Act 2014.* This Act legalised prolonged detention and forced return at sea, while also expunging Australia's obligation to international law, specifically relating to *non-refoulement.* These changes to domestic law not only undermined the Refugee Convention and Protocol, they in fact directly opposed it. These changes were made, however, in order to retroactively legalise this event of detention at sea, specifically addressing the length of time of this detention. Clarifications were made to the *Maritime Powers Act* in order to emphasise the need to afford officers of migration governance flexibility in their control of migrants in maritime spaces. An Explanatory Memorandum stipulates that "Parliament's intent is that this is a broad provision which provides maritime officers with the flexibility and discretion needed to effectively exercise maritime powers in real-world operational circumstances".[3]

Several months after the detention at sea, the case was heard in October 2014 at the High Court of Australia in Canberra. This was a momentous case determining how Australia's *Maritime Powers Act* should be applied at sea. The High Court ruled by a majority of 4:3 that the detention of asylum seekers for the period of one month was not, at any time, unlawful. The High Court also recongised that these actions were permitted by the *Maritime Powers Act*, meaning they did not have to dispute whether or not it was sanctioned by the government's non-statutory executive powers. The majority held that the asylum seekers could effectively be interdicted in the contiguous waters of Australia and removed to a "place" beyond Australia. This was influenced by the Amendment to the *Maritime Powers Act* [4] made by the Commonwealth Parliament months after the detention of the Plaintiff and instituted to retrospectively facilitate the detention of the Tamil asylum seekers (Tomasi, 2015, p. 428).

---

[3] Clause 3, para 13, Migration and Maritime Powers Legislation Amendment (Resolving the Asylum Legacy Caseload) Bill, 2013-2014, House of Representatives, Explanatory Memorandum.

[4] This is referring to the *Migration and Maritime Powers Legislation Amendment (Resolving the Asylum Legacy Caseload) Act* (2014).

The case orientated around whether Australia's officers were authorised to carry out such an extended period of detention at sea. The Commonwealth maintained that the *Maritime Powers Act* (MPA) or the non-statutory Commonwealth executive power legitimated the extended detention of the plaintiff. Section 72(4) of the MPA was central to the debate.[5] It reads as follows:

> A maritime officer may detain the person and take the person, or cause the person to be taken:
>> (a) to a place in the migration zone; or
>> (b) to a place outside the migration zone, including a place outside Australia.

The plaintiff argued that S 72(4) had to be understood in the context of Australia's obligation to international law, specifically that of *non-refoulement*. This would mean that a migrant who is intercepted, contained and removed at sea could not be taken to a place in which they face persecution, or to a place that is not signatory to the Refugee Convention and which does not protect against onward *refoulement*. Furthermore, the plaintiff also argued that if a decision on the destination is not made before departure, the period of time in which a person can be detained in order to remove them to a space outside Australia can exceed what is reasonably justifiable. Without a pre-determined destination, detention at sea can become indefinite. As the plaintiff posited, "There must be a limit discernible on the time of detention which must be ascertainable and not in the discretion of the Commonwealth".[6] The plaintiff also disputed whether Australian authorities are authorised to effectuate removals without asylum procedures when interdicting migrants in the contiguous zone.

A central aspect of this case orientated around the terms "to detain" and "to take" in S 72(4) of the MPA. Indeed, two of the judges declared the terms "to detain and to take" to be "the central focus of this case".[7] Reflecting this, the phrase is repeated recursively in the ruling by the presiding judges. The judges collectively ruled that the terms "detain" and "take" need to

---

[5] This section was not changed through the amendment to the MPA in 2014.

[6] QC R Merkle *in CPCF v Minister for Immigration and Border Protection & Anor [2014] HCATrans 227* (14 October 2014).

[7] J Hayne and J Bell, Para 67, *CPCF v Minister for Immigration and Border Protection* [2015] HCA 1 28 January 2015 S169/2014.

be understood in conjunction. In other words, *detaining* a migrant and *taking* a migrant were determined not to be distinct actions at sea, but rather one continuous action. As Justice Gageler notes, "Detention of a person under the provision triggers a concomitant duty to take the detained person to a place.".[8] This interesting framing, linking detention to coerced mobility, was not a disputed aspect of this case, despite having a significant impact on practices at sea. In fact, two of the judges who ruled the maritime detention of the Sri Lankan asylum seekers to be unlawful *still* held that these terms were to be understood in conjugation: "The power given by S 72(4) to detain and take a person to a place outside Australia is understood better as a single composite power than as two separate powers capable of distinct exercise.".[9] The idea that "to detain is to take" is to some extent implicit. If someone is interdicted and detained at sea on a mobile Customs or naval vessel, at some point, a "taking" somewhere else is necessary. Yet, in the context of this case, this "taking" is more complex. As explained above, the "migration zone" has been expunged from the territory of Australia. It is thus a legal device rather than a place and all migrants arriving by sea without a valid visa are precluded from accessing this legal device. Hence, a maritime migrant can only realistically be taken somewhere outside Australia, or to an Australian territory for a temporary period of time before being taken elsewhere. In concluding that detaining requires a concomitant taking at sea, the judges thus sanctioned detention for the purpose of onward mobility in the form of removal. As the MPA relates specifically to maritime migration regulation, this conclusion by the court firmly established Australia's seascape as a geography that facilitates the carceral transportation of migrants.

Justice Gageler explained in relation to the detention and removal of asylum seekers at sea "… the place [to which they are removed] need not be a place which is *proximate* to the place of detention, and it need not be a place with which the detained person has *any existing connection*".[10] This statement is significant for a number of reasons. Firstly, it detracts from the obligation to disembark rescued persons at the nearest safe port or the "next port of call". There is no firm international law mandating where people are disembarked (van Berckel Smit, 2020). As UNHCR note, "The obligation to come to the aid of those in peril at sea is beyond doubt. There is, however, a lack of clarity, and possibly lacunae, in international

---

[8] Ibid., Para 376.

[9] Ibid., Para 90.

[10] Ibid., Para 377, emphasis added.

maritime law when it comes to determining the steps that follow once a vessel has taken people on board" (UNHCR, 2002 para. 11). Place of safety is also "ill-defined" in both the SOLAS and SAR conventions (van Berckel Smit, 2020, p. 506). The nearest safe port can include the following port where a vessel is travelling to, which is particularly relevant in the case of rescues performed by commercial vessels; the port closest to the rescue location; or a port that is considered better equipped to provide care and assistance to those onboard (Papastavridis, 2018). UNHCR note that safe port could also include returning those to their place of embarkation as it is the responsibility of states "to prevent un-seaworthy vessels from leaving its territory", so long as this does not constitute a *refoulement* (UNHCR, 2002 para. 30). Ultimately, "ensuring the safety and dignity of those rescued and of the crew, must be the overriding consideration in determining the point of disembarkation" (UNHCR, 2002 para. 30). Disembarking rescued migrants should thus be done in a time sensitive way to a place where they do not face persecution. In determining that the place to which a migrant should be removed "need not be proximate", J Gageler affirmed that the *Maritime Powers Act* need not take heed of these international recommendations, and can instead prolong journeys for the purpose of disembarkation at a distant geography. In light of this, "taking" does not lead to a disembarkation that is for the benefit of migrants or the vessels carrying them, rather "taking" denotes a removal to geographies beyond Australia, geographies which need neither be "proximate" or "convenient".

Secondly, the ambiguity of the destination was further emphasised by J Gageler, who articulated that it should also be flexible. J Gageler argues that there should be no restriction on the places chosen for disembarkation, referring in particular to territories where the Australian government may not have a prearranged agreement to disembark migrants. Requiring an agreement prior to travel would, according to J Gageler, introduce limitations to the MPA which do not presently exist within the Act: "Having regard to the myriad circumstances in which, and myriad geographical locations at which, the maritime power to detain and to take might fall to be exercised, it would amount to a significant constraint on operational flexibility.".[11] In suggesting that limiting destinations would restrict operational flexibility, J Gageler emphasises the centrality of mobility to the function of the *Maritime Powers Act*. This removes all restriction on the destination of these maritime mobilities, or

---

[11] Ibid., Para 379.

what J Keane repeatedly refers to as "compulsory movements".[12] Thus, the destination need not be "proximate", "convenient" or predetermined. Rather, it can be at any distance from Australia and be a territory subject to open negotiations. This renders all third-party territories potential spaces that migrants can be removed to via the sea. Here we see how "taking" becomes a profoundly impactful term, indicating an expansive form of movement without constriction. This encourages it to become a violent term, one that suggests endless maritime journeys that keep migrants in a condition of partial rights, en route to uncertain destinations.

As an effect of such lengthy journeys that emerge from having variable and distant destinations, these statements imply that the length of time a migrant is detained at sea is not of significant concern. If migrants are taken across vast maritime geographies rather than disembarked at the nearest safe port, the period of time in which they are detained is unnecessarily prolonged, as was the case with the detained Tamil asylum seekers. Thus, through prioritising the "myriad circumstances and locations" of detention and mobility, J Gageler's statement legitimates the indefinite detention of migrants at sea. This indefinite detention is affirmed by another judge, J Crennan, who declared, it "is not necessary to ensure respect for the plaintiff's personal liberty or to avoid indefinite detention or detention at the discretion or whim of the Executive government.".[13] As such, in the *CPFC* case, "to detain is to take" is articulated to be a complex phrase which permits endless maritime mobility as well as indefinite detention at sea. While Australia authorised indefinite detention on territorial geographies under the Keating government in 1992, turning the maritime geography into zone within which migrants can equally be indefinitely detained marks a profound development in Australia's detention policy.

In the *CPCF* court case, three of the judges ruled in favour of the plaintiff, concluding that detention for the purpose of removal to India was not a lawful application of the *Maritime Powers Act*. Two of these judges were J Hayne and J Bell, who stated "What is presently important is that the power is to take to 'a place', not 'any place', outside Australia. The use of the expression 'a place' connotes both singularity and identification."[14] The attempt to

---

[12] Ibid., Para 424.

[13] Ibid. Para 218.

[14] Ibid. Para 92.

remove the asylum seekers to various undefined geographies was thus considered by J Hayne and J Bell to be unjust. Nonetheless, none of the three dissenting judges disputed that *detaining* a migrant demanded a *taking* of that migrant somewhere else. This is significant as maritime arrivals have no right to claim asylum in Australia, thus a *taking* implies a *taking away*. Moreover, it is important to consider than in the dissent by these judges, a number of significant points raised by the plaintiff were not responded to. There was no acknowledgement of Australia's duty to protect against *refoulement* and the extra-territorial obligations of the Refugee Convention, although there was recognition by CJ French and J Crennan that such obligations may have extraterritorial effects. In fact, the judges did not engage in a detailed way with international refugee law (Kaldor Centre for International Refugee Law, 2015a). Additionally, the conditions of detention were not challenged, despite reports of medications being confiscated, families separated, and a restricted access to fresh air onboard the vessel. As such, the minority who held this detention was unlawful did not contest the broader way this Act contravenes international migration law.

In the rulings made by the presiding judges, there are several significant statements which highlight the significance of the maritime geography to the justification of this at sea detention. In discussing the various on-ward mobilities that occurred at sea during the detention of the Tamil asylum seekers, J Keane states that such continuous mobility "is hardly surprising given the unpredictability of the circumstances of such voyages".[15] There was, however, no unforeseeable event that prolonged this detention: there was no shipwreck after the embarkation of the asylum seekers on the Customs vessel, nor was there any unforeseen maritime event. The one thing that did prolong the nature of the journey was India refusing disembarkation, which is precisely what the plaintiff was arguing led to an unfairly extended detention and what in fact the judges deemed to be acceptable. In emphasising the "unpredictable" nature of maritime voyages J Keane, who ruled this detention to be lawful, drew to the fore vague assumptions of the maritime as a mutable space that is erratic and unruly. J Crennan also emphasised the maritime geography as demanding flexibility in operations, quoting a former reading of the Maritime Powers Bill (2012) "The unique aspects of the maritime environment merit a tailored approach to maritime powers, helping to ensure flexibility in their exercise and to assist maritime officers to deal with quickly changing

---

[15] Ibid. Para 478.

circumstances and often difficult and dangerous situations.".[16] These statements suggests that the prolonged at sea detention was a consequence of the unpredictability of the maritime geography. This language thus entangles the maritime in the justification of indefinite detention at sea, suggesting that it is as responsible in prolonging coerced mobilities as the operatives arbitrarily moving migrants between indeterminate destinations.[17]

Within these statements made by the judges, the geography of the maritime is tied into the justification of detention and removal. In asserting that that unconstrained mobility at sea is operationally pivotal to the *Maritime Powers Act*, J Gageler emphasised that the space of maritime should be used *to keep migrants mobile*. This implicitly frames the sea as a surface of transportation. The maritime geography has long been idealised a "friction-free transportation surface" in which goods can be transported internationally without impediment (Steinberg, 2001, p. 125). Yet, in ruling that "to detain is to take" in migration governance, this case centralises the function of the sea as a space of transportation in the control of migrants. Moreover, in emphasising the mutability of the maritime as causational to the length of immigration detention at sea, the judges further amplified the significance of the maritime to carceral mobility at sea, using it to justify extended periods of confinement. Thus, the maritime was profoundly entangled in the ruling of this detention at sea as lawful, with assumptions of this geography as a space of transportation with a capricious nature used to legitimate indefinite detention and onward mobilities. Through this event of prolonged detention and the High Court case that came to justify the practices of the Australian authorities as necessary due to the condition of the maritime environment, the agenda of migration control in Australia developed from one premised on coerced mobilities, to one that sanctioned indefinite detention to facilitate potentially endless mobility.

Mobility as a method of containing migration is not unique to Australia. In the proliferation of mobilities at the Greek and Turkish border, migrants have not only testified to violent and life-threatening tactics of these coerced mobilities, where they are left in maritime locations outside Greek territorial waters, or Turkish territorial geographies. They have also revealed how in many cases, they had already reached land-based geographies of Greece where they

---

[16] Ibid. Para 201.

[17] It should be noted that Keane had a commitment to facilitating returns at sea. In relation to *non-refoulement*, Keane declared that "Australian courts are bound to apply Australian statute law 'even if that law should violate a rule of international law'." (Para 462).

were detained without access to asylum procedures and then later returned by sea (Pro Asyl, 2013, p. 10). This mobility also proliferates within Europe with the coercive relocation of migrants within France and Italy (Tazzioli, 2020). In these instances of migration governance, control emerges through a strategy of keeping mobile. The coerced mobilities informing migration governance are thus not distinct to the Australian context. What is specific about the Australian context is the extent to which this coerced mobility is explicitly ratified in domestic law as central to the state's strategy of migration control at sea. Equally, detention for prolonged periods of time at sea is not unique to Australia but rather reflects a growing inclination to turn maritime geographies into carceral spaces for mobile populations that states seek to regulate and discipline. In 2017, the *NY Times* reported on the US' "floating Guantánamos" and the US Coast Guard vessels that travel thousands of miles from US territories to interdict and detain drug smugglers at sea (Wessler, 2017). These detentions transpire under Operation Martillo, initiated in 2012, which operates between South and Central America, interdicting drug smuggler vessels and bringing those detained to trial in US courts. In the context of indefinite maritime detention in Australia, the agenda is in fact the opposite, indefinite detention at sea enables myriad onward mobilities until migrants are disembarked in anywhere *but* Australia.

This event of prolonged detention at sea and the indefinite detention in Australia's maritime environment that was later sanctioned by the High Court elucidates the carceral reworking of Australia's maritime migration governance strategy. Carceral studies scholars have explored the constant mobility that informs carceral systems, including the cyclical movement of visitors, staff, techniques and technologies, and prisoners between and beyond formal penal sites (Gill *et al.*, 2018; Mincke, 2020; Moran, 2016). The constancy of this movement has encouraged Mincke (2020, p. 7) to ask "whether the prison is best defined by its boundary". As Mincke articulates, the carceral can perhaps better be understood as "restricting or encouraging—the mobility of convicted offenders, and not as a territory isolated from the rest of society." (2020, p. 8). In this sense, the carceral does not produce stasis, but rather emerges in the myriad controlled and coerced mobilities. Focussing on these controlled mobilities shifts our focus away from the liminal geographies that denote a parameter, to consider the mobilities that exist as a method of governance. This leads us to ask a similar question of migration governance at sea: to what extent is it best defined by a border? If we consider the myriad mobilities that define Australia's control over migration at sea, the parameters of inside and outside lose pertinence, while the governance that transpires through mobility

appears definitional. In fact, in Australia, the maritime geography is not one in which the "inside" and "outside" of the state are easily discernible, as rights for migrants at sea have been annulled, while policing regimes at sea have been expanded (Dickson, 2021). Indeed, with Australia eliminating its migration zone, the "inside" of this state for maritime arrivals has become an elusive concept. In other words, the constriction of migratory movements at sea is, like the prison, not "based on *crossing*... [but is rather] understood through modification of the relative position of the points under consideration." (Mincke, 2020, p. 8). The geography of the maritime is pivotal to this, with its liquid spatiality used to justify movement as necessary, while its ever-changing surface allows such onward mobilities to be justified as indefinite in nature.

**Conclusion**

There is little transparency into events of interception and at sea mobilities under Australia's militarised migration governance programme, Operation Sovereign Borders. Through various reports and Parliamentary hearings, what is known is that between 2013 and 2021, 873 migrants endured coerced mobilities at sea which culminated in them either being returned to the place they were fleeing or taken to a third country (Refugee Council of Australia, 2021b). During this time, 1308 people were classified as "unable to be returned", however, these "unsuccessful returns" all occurred between 2013 and 2014, with none occurring since. Only 569 of these people are classified as being "temporarily in Australia", meaning they are within the state yet unable to settle there. The rest have either since been returned to their country of origin (364), resettled in a third country (253), or taken to an offshore site of immigration detention (116) (Refugee Council of Australia, 2021b). Australia's agenda of maritime migration governance is leading to a carceral reworking of maritime geographies. This carcerality emerges not only as a result of the indefinite detention that is now permissible at sea as a result of the *CPCF* case in 2015. It also emerges through the containment and control that is rendered through keeping migrants in a condition of prolonged mobility, where they suspended in a state of rightlessness and heightened policing. The judges of the *CPCF v Minister for Immigration* determined that under the Australian Maritime Powers Act, "to detain is to take" in migration governance at sea. Yet, as the judges detailed where and how this *taking* can occur at sea, the phrase acquired multiple meanings. "To detain is to take" denotes a *taking away* from Australia, as well as a *taking away* of autonomous mobility, the right to asylum, and protection against arbitrary detention and

*refoulement*. While these carceral mobilities transpire across maritime and terrestrial geographies, there is a specificity of the sea to Australia's agenda as it was through assumptions of the space of the maritime as unpredictable yet a space of transportation that the judges concluded that the destination of at sea mobilities should remain flexible, and as such, the detention of migrants at sea during these mobilities should exist without time constraint. The blurriness of this liquid geography has thus enabled a carceral mobility to emerge as central to the governance of migration at sea.

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
