# Peer review of "Immigration Detention at Sea: The carceral mobility that defines maritime migration governance in Australia"

_Migration Politics_

## Round 1 · Referee Report · Anonymous (Referee 1) · 2023-1-1

Strengths

1) The article offers an important detailed reading and review of “the containment and exclusion” that occurs at sea and in the geographical context of migration to Australia.

2) In conducting discursive analysis, the author carefully unpacks the various phases in Australia’s migration policy in streamlined prose. This kind of fine-tuned analysis is much needed to show the complex ways that policy decisions about migration get determined in legal spaces.

3) The article further goes beyond legal analysis to interrogate “liquid geographies” and “carceral mobilities” offering specific context and interesting theoretical lens to critique migration governance in Australia.

4) The abstract and conclusion are both well-written and make clear arguments around the language of “carceral mobility” in migration governance.

Weaknesses

1) The terminology around carcerality, mobility, “liquid” and “wet” geographies and the maritime as space could be further theorised. These terms situate the arguments in a specific literature which could be more fully referenced and reviewed.

2) While this may be a mere disciplinary preference, the reviewed literature in this topic is also somewhat unsatisfying in places. In many places the article employs either “serial citations” or long quotes, which could be revised to rather offer a clearer synthesis of what the literature on this topic argues.

3) While this is not an empirical article, could the experience of the Tamil asylum seekers be better included, even from news sites or reports? This may enhance the author’s argument about the ‘violence’ and the experience of carceral mobility.

Report

I enjoyed and learned a lot from reading this important article, which critically details the language of legal debates that have shaped Australia’s maritime migration governance. The detailed research presents insightful analysis of the language contained in legal documents, debates, and responses by judges. It describes the Australian approach to asylum seekers as one of “carceral mobilities” and of “taking away of autonomous mobility.” This involves migrants being sent back out to sea and moved from place to place for an indefinite time, sent to various “distant geographies,” or even sent back “home” - which the article argues defies Australia’s obligation to non-refoulement. The article strongly concludes that the “blurriness” of the maritime space “has thus enabled a carceral mobility to emerge” as a key part of Australia’s migration governance strategy.

The article meets the journal's acceptance criteria in that it offers a critique of migration governance from a law and social science perspective. It also offers a novel link between geography concepts and legal debates.

I have listed what I believe are the strengths and weaknesses of the article. In this report I expand on these and offer possible suggestions for further thought :

1) Some of the authors cited (Moran and Gill) have written about the agency of migrants in carceral spaces. I mention this only because some of their other work is cited, but the experience and perspective of the Tamil asylum seekers is not included here. There is a brief mention in the article of how they were denied medicine and sunshine; I would have liked to read more about their circumstances at sea. Would it help to also offer a few lines of historical context on the human rights violations in Sri Lanka?

2) May I ask whether the more recent work by Steinberg and Peters on "wet ontologies" may be useful in the literature review.

3) I wonder how the discussion of detaining and taking in shaping Australian migration governance relate to the literature critiquing the notion of deterrence (I recall the work of Pickering and Weber on “deterrence scripts” and Jason De Leon’s critique of “deterrence” in the US context for example)?

4) Would the article benefit from situating itself within some of the recent critiques of Western-centric migration governance (such as Anna Triandafyllidou?)

Stylistically:

1) In the introduction one finds repetition where the author ends many paragraphs with a repeat of their argument.

2) The last section is quite lengthy and perhaps subsections would help to guide the reader on the themes that appear in the analyses.

3) Some references have either unnecessary or incomplete information.

Requested changes

1) Revise the literature review to better synthesise key debates and theoretical concepts used in the article.

2) Even if brief, provide more context to the situation of the Tamil asylum seekers.

2) Delete repetition and add subsections to the lengthy sections .

3) Check reference list for errors.

---

## Round 1 · Referee Report · Anonymous (Referee 2) · 2023-1-12

Strengths

The paper is well written.
The topic is compelling and the case being examined is significant for migration studies -- mobility as governance is also a significant topic to be explored and has a significant impact on many maritime geographies.

Weaknesses

The opening of the paper is perhaps too long, or the paper needs to be reorganized more generally. In the first six pages, the author often signposts the significance without any analysis. I think the piece should move more quickly into analysis -- showing rather than telling why CPCF v. Minister for Immigration and Border Protection & Anor is a significant case for thinking mobility as governance and carceral geographies.

Report

The opening of the paper is perhaps too long, or the paper needs to be reorganized more generally. In the first six pages, the author often signposts the significance or focus of the paper, but I think the piece should move more quickly into analysis -- showing rather than telling why CPCF v. Minister for Immigration and Border Protection & Anor is a significant case for thinking mobility as governance.

The major interventions of the piece need to be honed in on more clearly: coerced mobility as governance, extending the carceral landscape, the significance of Australia legalizing this (in the context of other internationally migrant policies). The argument gets lost in the description of the case and the signposting -- I would like to see the theoretical intervention guide the structuring of the piece.

Requested changes

  1. I would recommend the paper be reorganized before being published. A restructuring would strengthen the argument, which is less compelling with the current introduction. Include a bit of textual analysis to frame significance.

  2. Include a clear depiction of how this example of coerced mobility is extending the carceral space in Australia (or more generally) -- what is the existing landscape? How does mobility redefine an understanding of the carceral? The author gets to this somewhat near the end of the article, but the argument would benefit from this being foregrounded and explained slowly and clearly.

  3. Phrases like "This statement is significant for a number of reasons." Can be removed or reworked to emphasize showing the significance rather than telling the reader something is important.

---

## Round 1 · Referee Report · Alessandro Corso (Referee 3) · 2023-3-1

Strengths

1 -There is good use of the relevant literature on returns, pushbacks and pullbacks, which demonstrates knowledge around the recent border regime dynamics across the world.

2-The historical context is important and it helps the reader to trace back the roots of present forms of forced migration management in Australia.

Weaknesses

1- the space that occurs between detention and mobility is only addressed vaguely. It feels as if ethnographic instances and/or more detailed descriptions of the experience of being in these liminal conditions may significantly improve the quality of the discussion.

2 - The style of writing is not always engaging and this risks to compromise the compelling argument of the author.

3 - Repetitions on the question of what detain and take mean within the context of surveilled irregular border crossing in Australia and elsewhere weaken the flow of the writing.

4 –

Report

The argument of this paper is timely, and it helps to expand the body of work on border regime strategies and its effects on mobility. By interrogating the procedures of the Maritime migration governance in Australia on the specific case of extended detention of Tamil asylum seekers from Sri Lanka at sea in June 2014, the paper successfully argues that forms of mobile detention have become strategies to legitimise indefinite detention. The border, the paper argues, emerges not purely as a space of inclusion and exclusion but as a site of containment through mobility.

Considering the suggested changes carefully, I recommend the paper to be published.

Requested changes

1-I would recommend the author not to push too far the inside/outside discussion but only say it once and convincingly (See pag 3).

2-I wonder whether the author may further explore the space that occurs between detention and mobility. To detain implies the responsibility to take those who are in a state of detention somewhere. The author well explains the issue and interrogates this process and the decision of where the detained migrants will be sent to. It would be interesting to also be aware of what happens while that decision is made. To detain is to take (away), but what happens before being taken away? What are the conditions of detention? Do people suffer? Do they die? Who is accountable for that? See past years episodes of migrants detained on rescue boats in the Mediterranean Sea who lost their life after delayed transfer to hospital.

3 -I would suggest integrating the recent article from Paolo Cuttitta titled “Over Land and Sea: NGOs\CSOs and EU Border Externalisation Along the Central Mediterranean Route” for the last section and the discussion on mobility and the discussion on inside and outside which may be well linked with the ongoing externalisation of borders.

4 -a closer engagement with the work of Maurizio Albahari on the Mediterranean region with the specific notion of “crimes of peace” and Ghassan Hage on waiting and unbearable life within the Australian context, may help to strengthen the analysis and provide insightful key concepts to consider.

---

## Round 2 · Author Response

Below, I respond in detail to the suggestions by the reviewers. I hope my comments are easy to follow and that my amendments meet the editors’ expectations.
Thank you very much.
Kind regards,
Andonea J Dickson

---

## Round 2 · List of Changes

Thank you to the anonymous reviewers for their helpful and constructive feedback. I have revised the article pertaining to the comments. Overall, the article has been re-written and re-structured as well as given a new and clearer title. I have created a stronger line of argumentation throughout with additional signposting which, in addition to edits to my writing, addresses concerns relating to the lack of engaging nature of the piece (reviewer 3). Information and data have also been updated, specifically in relation to the number of asylum seekers who have been subject to at sea mobilities as a result of Australian migration governance (page 10-11 and page 21). Below are more specific details of the re-writing, structured by the sections of the article and pertaining to requested changes from reviewers.
Section one:
In addressing suggestion 1 by reviewer 2, I have re-written and re-structured the first section of this article. This section now provides a clearer theoretical outline, demonstrating my focus on coerced mobility at sea and how this practice expands the carceral landscape of Australian migration control. This is now also more clearly signposted throughout the article, creating a clearer structure throughout.
The re-writing of this section means it reads less like a literature review with serial citations, and rather emphasizes my analysis upfront and the significance of mobility to carcerality (addressing a weakness identified by reviewer 2 as well as suggestion 2 by reviewer 2).
I have diminished reference to the phrasing “to detain is to take”, a weakness identified by reviewer 3 as disrupting the flow of the section.
In response to suggestion 2 by reviewer 1, I have ensured there is not a repetition of my argument at the end of sentences.
In response to a weakness identified by reviewer 2, that the opening of the paper is overly long, the re-writing of this section has not only created greater clarity in argument but has also reduced the word count of this section by approx. 400 words.
Correlating to a weakness identified by reviewer 1, my use of terminology is now clearer. However, it felt tangential in this particular article to provide further detail on the carceral wet or liquid geographies, as this is the topic of a former article (Dickson 2021).
Section two:
Addressing suggestion 2 by reviewer 3, on the space between detention and mobility, I have included a paragraph on page 12 detailing the experience of Tamil asylum seekers upon the vessel, as well as evidence from the judges demonstrating that this confinement was an explicit case of a detention. This provides a clearer connection between mobility and detention (which I also outlined more clearly in section 1).
This change also responds to suggestion 2 by reviewer 1, adding detail to the experience of the Tamil asylum seekers. I wholeheartedly agree that the inclusion of further testimonies would be interesting and advantageous. Yet, this would require further fieldwork and is beyond the scope of these revisions.
In response to reviewer 1’s comments, I have included reference to the agenda of “deterrence” in migration governance on page 9, with reference to De Leon (2015) Pickering and Weber (2014) and Matera et al (2023).
Pertaining to suggestion 3 by reviewer 3, I have drawn reference to the interesting articulation of mobilities within Paola Cuttitta’s article on page 4. However, as the case I am examining relies exclusively upon state-based actors operating within and just beyond Australian territories, I don’t draw on Cuttitta’s reference to externalization.
Section three:
Pertaining to suggestion 3 by reviewer 1, I have added subsections to the third section of the article. These subtitles are ‘To Detain is to Take’ (the original title of the article) and ‘A Geography that Demands Carceral Mobility?’, and I believe they make this section more readable as well as more clearer signpost my argument.
Pertaining to suggestion 1 by reviewer 3, I have diminished reference to the inside/outside discussion, with reference to it only on page 3 and not in the third section.
Overall:
There were some longer paragraphs across the article that have been edited to read more succinctly (this applies especially to the re-writing of the 1st section of the paper). I have also edited or deleted phrases that weaken the writing, such as “this statement is significant for a number of reasons”, suggestion 3 by reviewer 2.
Reference to the indefinite nature of detention at sea has also been refined, adding more detail and clarity to this ruling and when and how time limitations apply.
I have reviewed the references – suggestion 3 by reviewer 1.

---

## Editorial Decision

unknown